# Towards Human Inverse Dynamics from Real Images: A Dataset and Benchmark for Joint Torque Estimation

## Abstract

Human inverse dynamics is an important technique for analyzing human motion. Previous studies have typically estimated joint torques from joint pose images, marker coordinates, or EMG signals, which severely limit their applicability in real-world scenarios. In this work, we aim to directly predict joint torques during human movements from real human images. To address this gap, we present the vision-based inverse dynamics dataset (VID), the first dataset tailored for the joint torque prediction from real human images. VID comprises 63,369 frames of synchronized monocular images, kinematic data, and dynamic data of real human subjects. All data are carefully synchronized, refined, and manually validated to ensure high quality. In addition, we introduce a comprehensive benchmark for the vision-based inverse dynamics of real human images, consisting of a new baseline method and a new evaluation criteria with three levels of difficulty: (i) overall joint torque estimation, (ii) joint-specific analysis, and (iii) action-specific prediction. We further compare the baseline result of our VID-Network with other representative approaches, our baseline method achieves the state-of-the-art performance on almost all the evaluation criteria. By releasing VID and the accompanying evaluation protocol, we aim to establish a foundation for advancing biomechanics from real human images and to facilitate the exploration of new approaches for human inverse dynamics in unconstrained environments.

## 1 Introduction

Human inverse dynamics is the process of computing the internal joint torques and forces required to produce a given human motion, based on observed kinematics and external forces. It encompasses a wide range of application domains, including medicine, sports, robotics, and rehabilitation (LeVeau, 2024). Representative works include the analysis of athletic movement techniques(Johnson & Ballard, 2014; Yeadon et al., 2006; Lech et al., 2015), the surgical replacement of damaged joints with prosthetic implants(Kameni Nteutse & Geletu, 2024; STEINER et al., 1989; Hu et al., 2024), and the study of motion control strategies in humanoid robots(Koonce et al., 2011; Sulaiman et al., 2024; Liang et al., 2024; Sy Horng Ting et al., 2025). Joint torque is a key element in biomechanical research, as it characterizes the mechanical interactions underlying human movement.

Existing approaches for torque estimation generally fall into three categories: **surface electromyography (sEMG)-based methods** (Buchanan et al., 2005; Paquin & Power, 2018; Gui et al., 2019; Caulcrick et al., 2021), **inverse dynamics (ID)-based methods** (Manukian et al., 2023; Johnson & Ballard, 2014; Xiong et al., 2019; Zell & Rosenhahn, 2017), and **imitation learning methods**(Liu et al., 2024; Luo et al., 2023; Peng et al., 2022; 2021). The first relies on sEMG devices to capture muscle electrical activity, which is then used as input to a neural network for torque prediction, or alternatively, is processed through a forward dynamics model. The ID-based method requires the collection of motion capture data, typically obtained via optical motion capture systems in conjunction with force plates. These data are then used within Newton-Euler dynamics formulations or data-driven models to estimate joint torques(Khalil, 2010; Riemer & Hsiao-Wecksler, 2008). Both sEMG- and ID-based methods are constrained to laboratory environments due to their dependence on specialized and expensive equipment, making them impractical for use in outdoor sports or competitive settings (Zhang et al., 2023; 2021b). To address these limitations, imitation-based meth-

ods have been proposed. These approaches leverage humanoid robot-based simulations to generate paired kinematic and dynamics data, which are then used to train neural networks to learn the underlying relationship(e.g., Inverse Dynamics) between motion and joint torques. While such methods have shown promising results, they suffer from inherent distributional discrepancies between simulated and real-world data, and still rely on marker-based motion input, limiting their applicability in unconstrained environments. **To overcome these challenges, we aim to develop a purely real human image-based solution for joint torque estimation that eliminates the need for marker entities and enables practical deployment in real-world or wild exercise scenarios.**

The first issue that needs to be addressed is the dataset. Currently, no data can be directly used for vision biomechanics inference. We derive the VID dataset from the open-source dataset (Uhlrich et al., 2023). The dynamics data were exported from the OpenSim software. We devoted substantial effort to synchronizing kinematic and dynamic frame data and refining dataset quality, ultimately providing 63,369 frames of real human images along with corresponding kinematic and dynamic annotations.

Secondly, we propose a baseline network(VID Network), which is designed to estimate joint torques purely from visual input, without relying on motion capture or force data. Since joint torque is inherently dependent on joint position, capturing accurate spatial structures is essential for reliable prediction. Given the impressive performance of existing CNN networks in 3D human pose estimation, we construct the auxiliary 3D pose estimator and the spatial probabilistic model. In the first training stage, these models were pre-trained on several large-scale 3D pose datasets. Based on the pre-trained model, we design the marker regressor and the TorqueInferNet. The marker regressor enables the network to learn the joint poses of interest. From multiple frames, the TorqueInferNet integrates spatial probabilistic features with the selected markers' position of each frame to predict joint torques. Extensive experiments show that our method has exceeded the state-of-the-art methods based on marker points. This demonstrates that this torque prediction solution based on real images is feasible. The main contribution of this paper is summarized as follows:

- **Dataset**: We introduce VID, a high-quality and carefully synchronized biomechanical dataset comprising 63,369 frames of real human images with corresponding kinematic and dynamic annotations, which can be directly used for vision-based joint torque prediction.

- **Benchmark**: We establish the first benchmark for torque prediction from real human images, including a comprehensive evaluation protocol with three levels of criteria: (i) overall joint torque estimation, (ii) joint-specific analysis, and (iii) action-specific prediction. This provides a standardized basis for fair comparison across future methods.

- **Baseline**: We propose VID-Network, a strong baseline model that integrates spatial probabilistic features, marker regression, and temporal modeling, achieving state-of-the-art performance and validating the feasibility of torque estimation directly from real human images.

## 2 RELATED WORKS

**Newton-Euler formulation**. It is the traditional method to solve the torque calculation, where generalized coordinates are used to describe the motion of a mechanical system. Arian (Arian et al., 2018) focused on analyzing the kinematics and dynamics of a special 3-DOF parallel robot called Tripteron by modifying its structure and using the Newton-Euler method. Luca (De Luca & Ferrajoli, 2009) introduced an improved Newton-Euler algorithm to make dynamics calculations easier and more effective for robot fault detection and control. The formulation is given by:

$$M(q)\ddot{q} + C(q, \dot{q}) + G(q) = J\lambda + \tau \qquad (1)$$

where the vector $\mathbf{q}$ denotes the generalized coordinates (e.g., joint angles), while $\dot{\mathbf{q}}$ and $\ddot{\mathbf{q}}$ represent their first and second derivatives, corresponding to joint velocities and accelerations. The matrix $M(\mathbf{q})$ is the mass or inertia matrix that describes how the system's mass is distributed. $C(\mathbf{q}, \dot{\mathbf{q}})$ accounts for Coriolis and centrifugal effects due to movement. $\mathbf{G}(\mathbf{q})$ represents gravitational forces acting on the system. On the right-hand side, $\boldsymbol{\tau}$ is the vector of applied joint torques or forces, and $\mathbf{J}\boldsymbol{\lambda}$ captures the contribution of external or constraint forces, where $\mathbf{J}$ is the Jacobian matrix and $\boldsymbol{\lambda}$ is the vector of Lagrange multipliers representing those constraint forces. This method requires

the use of a motion capture system to obtain the joint pose $\mathbf{q}$, from which joint velocities $\dot{\mathbf{q}}$ and accelerations $\ddot{\mathbf{q}}$ are computed via numerical differentiation. Ground reaction forces are collected using a force plate, and a human musculoskeletal model with parameters such as mass, inertia, and linkage structure is constructed using OpenSim(Delp et al., 2007). Owing to limitations such as equipment, location, and duration of data collection, this method cannot be quickly applied to real-time tasks.

**Deep Learning Methods**. The rapid development of machine learning methods has significantly advanced the prediction of dynamics systems. Machine learning methods can be employed to predict human biomechanics using the information collected by sensors, including sEMG data, keypoint positions of pose, force platform reactions, and so on. For example, (Zhang et al., 2021a) proposed an electromyography (sEMG) driven neuromuscular skeletal (NMS) model and an artificial neural network (ANN) model for estimating ankle joint torque. ANN models perform better when the training data contains a large and diverse range of motion types. Some (Son et al., 2024; Zhang et al., 2020; Wang et al., 2023; Zhang et al., 2022) used Long Short Term Memory (LSTM) neural networks and transfer learning to predict lower limb joint torque, which is applicable to various scenarios in daily activities and provides new ideas for the application of wearable devices in motion analysis and rehabilitation. (Dinovitzer et al., 2023) proposed a hybrid method combining neural networks and dynamics models, as well as an end-to-end neural network, for real-time estimation of human joint torque to dynamically predict human walking. The hybrid model showed high accuracy in simulated environments, while the end-to-end neural network performed better in actual testing. However, the hybrid model had better generalization ability in scenarios different from the training data. (Zell et al., 2020) addresses the problem of human dynamics estimation by proposing a weakly supervised learning framework. The core idea of the framework is to leverage easily accessible motion data and employ weak supervision and domain adaptation to estimate ground reaction forces, ground reaction moments, and joint torques.

**Motion Imitation Learning**. It refers to the process where an agent learns to replicate human or expert motion trajectories by observing demonstration data, typically in the form of joint positions, velocities, or full-body kinematics. (Kobayashi et al., 2025) introduced a new Transformer model-based imitation learning method (ILBiT) for autonomous operation of robot arms. (Matsuura et al., 2023) proposed a study on imitation learning for humanoid robots, focusing on solving the development problems of teleoperation equipment and high load control systems. Based on the data obtained from imitation learning to train neural networks, recent work such as ImDy (Liu et al., 2024) has collected up to 150 hours of data using this method, processed the input motion state sequence using Transformer encoders, and predicted joint torque and ground reaction force through linear head prediction. The advantage of this method is that it can easily collect motion data of various actions and durations, but the disadvantage is that the inherent differences between imitation learning and real motion pose challenges to model generalization.

# 3 VISION INVERSE DYNAMICS DATASETS

As noted previously, the majority of current datasets are not well-suited for torque prediction from real human images. One reason for this is that processing motion data using biomechanical modeling software demands significant manual effort. Additionally, there is the challenge of synchronizing data from various sources. Some datasets (Zell et al., 2020; Werling et al., 2024) have kinematic and dynamics data and pose images, but lack real images; Some datasets (Uhlrich et al., 2023; Mahmood et al., 2019) only have real human images and kinematic data, without synchronized dynamics data or high-quality data. In this work, we present an optimized dataset derived from open-source datasets (Uhlrich et al., 2023; Mahmood et al., 2019), enabling end-to-end mapping from real images to biomechanical dynamics and facilitating future research in this field. The dataset is augmented by annotating joint velocities and torques, resulting in more complete kinematic and dynamic data. The comparative information is shown in the Table1 below. Ours have full kinematics data, dynamics data, and real images. All the data were manually synchronized and smoothed to remove outliers.

The dataset comprises recordings from 9 subjects (including 4 males and 5 females) with body heights ranging from 1.60 m to 1.85 m. Each subject performed seven types of movements, from

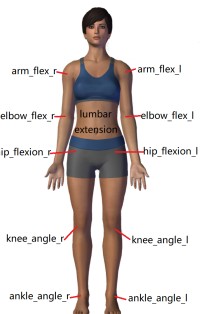

Figure 1: Labeled Anatomical Locations of Joint Used for Torque Evaluation.

| Dataset Name | Size | Kinematics | Dynamics | Sync. | Real Img. |
|---|---|---|---|---|---|
| CMU Mocap (CMU, 2003) | 4.5h | Partial | Partial | ✗ | ✗ |
| AMASS (Mahmood et al., 2019) | 40h | Partial | Partial | ✗ | ✗ |
| OpenCap (Uhlrich et al., 2023) | 8h | Partial | Partial | ✗ | ✓ |
| Imdy (Liu et al., 2024) | 152h | Full | Full | ✓ | ✗ |
| AddBiomechanics (Werling et al., 2024) | 70h | Full | Full | ✓ | ✗ |
| **Ours** | 63,369f | Full | Full | ✓ | ✓ |

Table 1: Comparison of existing biomechanics-related datasets, including dataset size, completeness and synchronization of kinematic and dynamic data, and availability of real images. Here, **h** denotes hours and **f** denotes frames.

which approximately 100 consecutive frames per trial were extracted at a sampling rate of 100 FPS. A total of 51 markers were placed on each subject's body. Using the OpenSim software, 35 joint positions and corresponding joint torques were manually extracted for each frame. Joint angular velocities were computed using the Finite Difference method. Given a sequence of joint positions $\mathbf{x}_i \in \mathbb{R}^3$ at discrete time steps $t_i$, the joint velocity can be approximated using finite differences. The calculation formulation was

$$\mathbf{v}_i = \frac{\mathbf{x}_{i+1} - \mathbf{x}_{i-1}}{2\Delta t},$$

where $\Delta t = t_{i+1} - t_i$ is the time interval between frames. To ensure data quality, kinematic trajectories were smoothed using a Savitzky–Golay filter (Savitzky & Golay, 1964) (window size = 11 frames, polynomial order = 3). Outliers exceeding a velocity-based threshold were corrected by cubic spline interpolation, applied only to short gaps ($\leq 5$ frames) to preserve natural motion continuity. In total, the final dataset contains 63,369 frames of synchronized visual, kinematic, and dynamics annotations. Obviously, much personal bioinformation is also available, such as height, mass, and gender.

## 4 METHODS

With the collected VID dataset, we aim to address the human inverse dynamics in a full-supervised manner with a vision inverse dynamics network. In the first subsection, we first introduce the formulation of data-driven inverse dynamics. Then, the proposed VID Network is introduced in the second subsection. The overall pipeline of VID is illustrated in Figure 2.

### 4.1 FORMULATION

The vision inverse dynamics task can be illustrated as the following equation,

$$\tau^{m*t} = \text{VID}(I^t, Pos^{m*t}, Velc^{m*t}, H, M), \qquad (2)$$

where $\tau^{m*t}$ are the predicted $m$ number of joint torques at timestamp t, $I^t$ is the visual image of person at timestamp t, $Pos^{m*t}$ are the markers' position at timestamp t, $Velc^{m*t}$ are the markers' velocity at timestamp t, $H$ and $M$ are the height and mass of the subject. Since we propose a purely

visual approach, $I^t$ is the model's input, and all motion information except for $I^t$ can be used as supervisory signals.

## 4.2 VID NETWORK

**Baseline Architecture**. To construct an intuitive yet effective real image-based baseline network, we adopt a standard design paradigm commonly used in existing 3D human pose estimation networks. This paradigm usually consists of a backbone feature extractor(ResNet-101) and a pose estimator model. This paradigm has been proven effective in joint position estimation (Cheng et al., 2020; Fabbri et al., 2020; Kang & Lee, 2024) and can provide valuable joint space information for our joint torque prediction task. The backbone is based on a convolutional neural network, which has been widely validated as a strong performer in visual recognition tasks. The spatial probabilistic model consists of a series of deconvolutional layers followed by a 1×1 convolutional layer. Its output is a set of features, where each channel represents the spatial probability distribution of a specific joint in the image. In addition, a pose estimator maps the generated spatial features to the 3D coordinates of anatomical joints, while a marker regressor predicts the positions of external markers from the same spatial representation. The position of marker points is more flexible and conforms to the anatomical structure of joints, which is crucial for predicting joint torque. To estimate joint torque, we further designed TorqueInferNet to combine spatial features with predicted marker coordinates and use Transformer Encoder(head=8, dim=128) to extract multiple frames near the target frame for prediction.

**Spatial probabilistic model**. To predict 3D spatial probabilistic features for each joint, we design a lightweight head network composed of a series of deconvolutional layers followed by a final prediction layer. The input to the head network is a high-dimensional feature map of shape $[B, 2048, H, W]$ extracted by the backbone. The deconvolutional module consists of three stacked transposed convolutional layers, each with a kernel size of $4 \times 4$, stride 2, and padding 1. These layers progressively upsample the feature maps and reduce the channel dimension to 256. Each deconvolution is followed by a batch normalization layer and a ReLU activation function. Finally, a $1 \times 1$ convolution is applied to transform the output into a tensor of shape $[B, J \cdot D, H', W']$, where $J$ is the number of joints and $D$ is the depth dimension of the volumetric spatial features. This output is used to represent the 3D spatial likelihood of each joint.

**Pose estimator**. To obtain continuous 3D joint coordinates from the spatial probabilistic features, we adopt a differentiable soft-argmax operation (Luvizon et al., 2019). Given the predicted spatial probabilistic features of shape $[B, J, D, H, W]$, where $B$ is the batch size, $J$ is the number of joints, and $(D, H, W)$ denote the depth, height, and width dimensions respectively, we first flatten the spatial and depth dimensions and apply the softmax function along this axis:

$$S\tilde{P}F_{b,j} = \text{Softmax}\Big(\text{reshape}(SPF_{b,j}, [D \cdot H \cdot W])\Big), \qquad \forall b \in [1, B], \ j \in [1, J]$$

The normalized spatial probabilistic features $SPF_{b,j}$ are then reshaped back to $[D, H, W]$, and the expectation along each axis is computed by summing over the other two dimensions:

$$x_{b,j} = \sum_{w=1}^{W} w \cdot \sum_{d=1}^{D} \sum_{h=1}^{H} S\tilde{P}F_{b,j}(d, h, w) \tag{3}$$

$$y_{b,j} = \sum_{h=1}^{H} h \cdot \sum_{d=1}^{D} \sum_{w=1}^{W} S\tilde{P}F_{b,j}(d, h, w) \tag{4}$$

$$z_{b,j} = \sum_{d=1}^{D} d \cdot \sum_{h=1}^{H} \sum_{w=1}^{W} S\tilde{P}F_{b,j}(d, h, w) \tag{5}$$

The final 3D joint coordinates are obtained by concatenating the $x$, $y$, and $z$ components for each joint:

$$\mathbf{c}_{b,j} = [x_{b,j}, y_{b,j}, z_{b,j}] \tag{6}$$

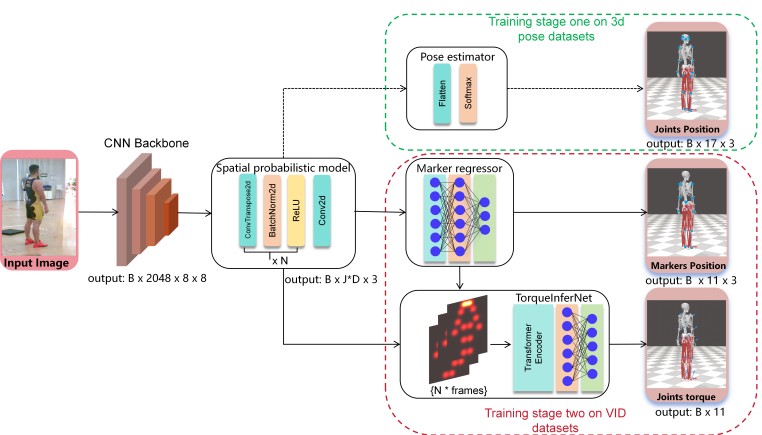

Figure 2: Our baseline network for joint torque prediction from real human images. In the first stage, the Pose estimator and spatial probabilistic model are pretrained on multiple large-scale 3D pose datasets. In the second stage, the TorqueInferNet integrates the joint position information predicted by the marker regressor and the spatial features encoded in the spatial probabilistic model, then use Transformer Encoder to extract multiple frames near the target frame for joint torques prediction.

This soft-argmax operation enables end-to-end learning and allows for sub-voxel localization precision, while preserving differentiability.

**Marker regressor**. To estimate the 3D positions of body-attached markers, we design a lightweight regression network that maps spatial probabilistic features to markers' coordinates. The network consists of a multilayer perceptron (MLP) with two hidden layers of 128 units, each followed by a ReLU activation. The output layer predicts the 3D positions of $M$ markers, resulting in an output of size $3M$. Formally, the network learns a function $f : \mathbb{R}^{B \times 21 \times 3} \to \mathbb{R}^{B \times M \times 3}$, where $M$ is the number of markers. The output is reshaped into a tensor with shape $[B, M, 3]$, which represents the predicted 3D coordinates for each marker.

**TorqueInferNet.** TorqueInferNet is a temporal regression network designed to predict joint torques from motion-related features using centered prediction. It takes both the flattened spatial probabilistic features and the 3D marker positions as input. Given a sequence of $T$ (default T=13)consecutive frames with features $\mathbf{x} \in \mathbb{R}^{B \times T \times N \times d}$, we first flatten and project each frame into a compact embedding of dimension $D$, yielding a token sequence $\mathbf{z} \in \mathbb{R}^{B \times T \times D}$. This sequence is processed by a Transformer encoder with multi-head self-attention to capture long-range dependencies across frames. The hidden state corresponding to the middle frame is then passed through fully connected layers to regress the joint torques at that time step: $f : \mathbb{R}^{T \times N \times d} \to \mathbb{R}^{J}$, where $J$ denotes the number of predicted joints.

**Loss terms**. In the first stage of training, we employ the *Mean Squared Error* loss to supervise the predicted 3D human pose against the ground truth annotations, denoted as $\mathcal{L}_{\text{pose}}$. In the second stage, we continue to use the MSE loss to minimize two objectives: the error between the predicted marker coordinates and the ground truth, denoted as $\mathcal{L}_{\text{marker}}$, and the error between the predicted joint torques and their ground truth values, denoted as $\mathcal{L}_{\text{torque}}$. The final loss used for optimization in the second stage is a *weighted sum* of $\mathcal{L}_{\text{marker}}$ and $\mathcal{L}_{\text{torque}}$, defined as:

$$\mathcal{L}_{\text{total}} = \lambda_1 \cdot \mathcal{L}_{\text{marker}} + \lambda_2 \cdot \mathcal{L}_{\text{torque}}, \tag{7}$$

where $\lambda_1$ and $\lambda_2$ are hyperparameters that balance the contributions of each term. And their sum is constrained to 1.

| Metric | Dino | Imdy | **Ours** |
|---|---|---|---|
| mPJE (N·m/kg) ↓ | 2.9493 | 2.9262 | 1.7612 |

Table 2: The overall performance on the VID dataset. Comparison of methods in terms of mean per-joint torque error (mPJE, N·m/kg). Lower is better.

| Joint Types | mPJE(N.m/kg) ↓ | | |
|---|---|---|---|
| | Dino | Imdy | **Ours** |
| hip_flexion_r | 0.4692 | 0.4914 | **0.2702** (-0.2212) |
| hip_flexion_l | 0.3888 | 0.3589 | **0.2432** (-0.1157) |
| lumbar_extension | 0.7811 | 0.8147 | **0.3326** (-0.4485) |
| knee_angle_r | 0.3952 | 0.4120 | **0.2044** (-0.1908) |
| knee_angle_l | 0.2765 | 0.2836 | **0.2124** (-0.0641) |
| arm_flex_r | 0.0464 | 0.0486 | **0.0307** (-0.0157) |
| arm_flex_l | 0.0698 | 0.0562 | **0.0405** (-0.0157) |
| ankle_angle_r | 0.2467 | 0.2088 | **0.1717** (-0.0371) |
| ankle_angle_l | 0.2333 | 0.1730 | **0.1234** (-0.0496) |
| elbow_flex_r | 0.0164 | 0.0362 | **0.0162** (-0.0002) |
| elbow_flex_l | 0.0269 | 0.0428 | **0.0204** (-0.0065) |

Table 3: Joint-specific mPJE (N·m/kg) on the VID dataset, showing that our method consistently achieves the lowest error.

## 5 EVALUATION

### 5.1 EVALUATION SETTINGS

To evaluate the effectiveness of the proposed VID network, we conducted extensive experiments. The compared methods include Dino (Dinovitzer et al., 2023), which is the best hybrid approach, and ImDy (Liu et al., 2024), which is the state-of-the-art imitation learning-based method. It should be noted that these two methods rely on labeled motion data to estimate joint torque, while our method relies on real images. The VID dataset is split into a training set and a testing set in an 8:2 ratio. Hyperparameters $\lambda_1$ and $\lambda_2$ were set as 0.5. All input images are resized to $256 \times 256$ pixels. We use the Adam optimizer with an initial learning rate of 0.001. The batch size is set to 32, and the models are trained for 500 epochs. During training, the learning rate is decayed to 0.0001 to ensure convergence and improved optimization performance. All methods were trained and evaluated using the same dataset configuration. The experiments were conducted on two NVIDIA A100 GPUs.

### 5.2 EVALUATION CRITERIA AND METRICS

We define three new evaluation criteria for the three methods in the experiment: **overall performance, joint-specific performance, and action-specific performance**. We adopt mean Per Joint Error (mPJE) as the evaluation metric. According to the design of the previous method (Liu et al., 2024), mPJE is further normalized by body weight to align different subjects. The specific calculation formula is shown in Formula8 below, where $N$ is the number of joints, $\hat{J}$ is the predicted joint torque, $J$ is the ground truth, $Mass_{sub}$ is the body weight of the subject.

$$\text{mPJE} = (\frac{1}{N} \sum_{i=1}^{N} \left\| \hat{\mathbf{J}}_i - \mathbf{J}_i \right\|_2 )/Mass_{sub} \tag{8}$$

### 5.3 EVALUATION RESULTS AND ANALYSIS

The experimental results of the three evaluation tasks are shown as follows: 1) The overall quantitative results are presented in Table 2. It refers to the average mPJE of all samples in the dataset, reflecting the overall performance of the models. Dino achieved an mPJE of 2.9493, while ImDy resulted in a lower mPJE of 2.9262. The proposed method yields the lowest error, achieving a mean Per Joint Error (mPJE) of 1.7612. This represents a reduction of 1.15 compared to the best-performing baseline, corresponding to a 39.81% relative improvement.

| Action Types | mPJE(N.m/kg) ↓ | | |
|---|---|---|---|
| | Dino | Imdy | **Ours** |
| SitToStand1 | 0.2266 | 0.2451 | **0.1252** (-0.1014) |
| STSweakLeg1 | 0.2714 | 0.2222 | **0.1389** (-0.0833) |
| squats1 | 0.2239 | 0.276 | **0.1405** (-0.0834) |
| squatsAsym1 | 0.222 | 0.2742 | **0.1422** (-0.0798) |
| walking1 | 0.3419 | 0.1722 | **0.1299** (-0.0423) |
| walking2 | 0.3120 | 0.1878 | **0.1436** (-0.0442) |
| walking3 | 0.3143 | 0.1780 | **0.1222** (-0.0558) |
| walkingTS1 | 0.3023 | **0.1657** | 0.1840 (+0.0183) |
| walkingTS2 | 0.2908 | **0.1546** | 0.2218 (+0.0672) |
| walkingTS3 | 0.2931 | 0.1609 | **0.1120** (-0.0489) |
| DownJump1 | 0.6511 | 0.6879 | **0.5341** (-0.1538) |
| DownJump4 | 0.6788 | 0.7305 | **0.5120** (-0.1668) |
| DownJump5 | 0.6569 | 0.7305 | **0.4926** (-0.1643) |
| DownJumpAsym3 | 0.9330 | 1.0242 | **0.7769** (-0.1561) |
| DownJumpAsym4 | 0.6421 | 0.7177 | **0.4412** (-0.2009) |
| DownJumpAsym5 | 0.6744 | 0.7207 | **0.4751** (-0.1993) |

Table 4: Action-specific mPJE (N·m/kg) on the VID dataset. Our method achieves the lowest error across most action categories. The numbers in parentheses represent the improvement compared to the best existing method.

2) To thoroughly evaluate the model's performance across different joints, we report the mean Per Joint Error (mPJE) for each joint. The quantitative results of joint-specific performance are shown in Table 3. The evaluation involves 11 types of joints, including: hip_flexion_r, hip_flexion_l, lumbar_extension, knee_angle_r, knee_angle_l, arm_flex_r, arm_flex_l, ankle_angle_r, ankle_angle_l, elbow_flex_r, and elbow_flex_l. The detailed positions of these joints are shown in Figure 1. IMDY and DINO demonstrate varying performance across the 11 joints, with each joint exhibiting its own specific advantages and disadvantages. However, the differences in their scores are not significant. Our methods got the better prediction results in all joint types. Excluding the elbow_flex_r and elbow_flex_l joints, our approach significantly outperforms the other two methods. The most notable improvement is observed in the lumbar_extension, with an enhancement of 0.4485, followed by a 0.2212 increase in hip_flexion_r.

3) The VID dataset comprises a total of 16 distinct actions, including transitioning from sitting to standing, walking, squatting, jumping down, and so on. To verify the performance of the model on different actions, we calculated the mPJE values for each action. The quantitative results of action-specific performance are shown in Table 4. Imdy significantly outperforms Dino on the walking action. However, its performance is relatively worse on other types of actions. We attribute this to the larger number of walking samples, which benefits transformer-based models—a phenomenon that has also been validated in previous studies. Our method still achieved the best performance on most action types, with a particularly notable advantage of up to 0.2009 on the DownJump category. However, its performance on walkingTS1 and walkingTS2 is inferior to that of Imdy. The performance of the three methods in the DownJump action is shown in Figure 3 below.

## 5.4 ABLATION STUDY

We designed an effective network architecture consisting of three main modules: the pre-trained pose estimator, the Marker regressor, and the TorqueInferNet. In general, directly connecting the TorqueInferNet to the spatial probabilistic features may represent a minimalist design paradigm. Therefore, we aim to investigate the impact of the two auxiliary modules(the pose estimator and the marker regressor) on joint torque prediction. The results of the ablation study are shown in the Table 5. The marker regressor, which extracts joint position information, has a positive impact on joint torque prediction. Similarly, the inclusion of the pose estimator leads to a performance improvement of 0.15, further validating our hypothesis that pre-training enhances the spatial representation capability of the backbone network.

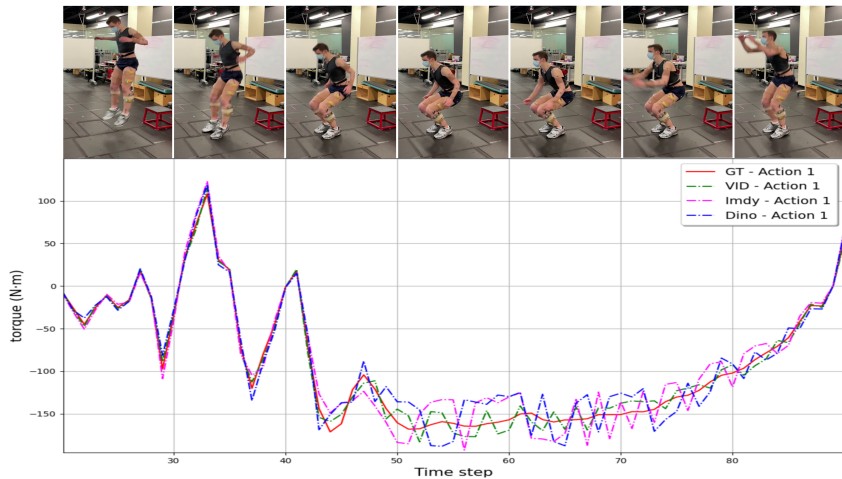

Figure 3: Visualization of the performance of three methods on the down jumping action.

| Ablation Settings | | | mPJE |
|---|---|---|---|
| Pose estimator | Marker regressor | TorqueInferNet | |
| ✗ | ✗ | ✓ | 2.4990 |
| ✗ | ✓ | ✓ | 2.1179 |
| ✓ | ✗ | ✓ | 2.2236 |
| ✓ | ✓ | ✓ | 1.7612 |

Table 5: Ablation Study on the Effectiveness of Auxiliary Modules.

# 6 CONCLUSION

In this paper, we propose an inverse dynamics prediction approach from real human images, which overcomes the limitations of previous methods in terms of application scenarios. We constructed the VID dataset by manually exporting and optimizing data from existing open-source datasets, and further augmented it with joint torque and velocity annotations. This manually synchronized dataset consists of 63,369 high-quality frames. Leveraging this resource, we develop a novel end-to-end neural framework, VID-Network. It comprises the Spatial Probabilistic Model for extracting spatial features of anatomical joints, the Marker Regressor for estimating the required joint coordinates, and TorqueInferNet, which effectively integrates spatial representations and positional cues to predict joint torques.

To better validate the effectiveness of the proposed approach, we introduce three evaluation criteria: (i) overall joint torque estimation, (ii) joint-specific estimation, and (iii) action-specific estimation. These criteria enable a more comprehensive comparison of model performance against existing methods. Compared to the previous methods, our method achieves a 39.81% improvement. It demonstrates that the proposed approach establishes a strong baseline for reference. To the best of our knowledge, this is the first study to estimate inverse dynamics directly from real human images, thereby establishing a new benchmark for this task. More importantly, it paves the way for applying torque prediction in more unconstrained and practical scenarios.

Although this paper provides the first benchmark for inverse dynamics based on real human images, it also opens several avenues for future research. Promising directions include: (i)cross-subject generalization, evaluating models on unseen subjects; (ii)in-the-wild scenarios, extending evaluation beyond laboratory settings; and (iii)multi-modal extensions, incorporating complementary signals such as IMU or EMG. We expect VID to serve as a foundation for these future benchmarks, stimulating broader progress in biomechanics and computer vision.

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

# A  APPENDIX

## A.1  EXTENSIBILITY AND REPRODUCIBILITY

Our goal in designing VID is to maintain high scalability and reproducibility. This means supporting expansion of the number and types of predicted joints, maintaining universal code design, and consistent evaluation benchmarks, facilitating the sharing of research results among researchers. We also designed a suite of tools for extracting joint kinematic and dynamic data, allowing researchers to focus on data preprocessing. All tools are implemented in Python, making them easy to deploy and run. The baseline model design also leverages many fundamental models, such as ResNet-101 and the Transformer Encoder. We ensured that the model's dependencies are easily installed, making it easier for researchers to replicate our results.

## A.2  DATASET

The VID dataset comprises 9 subjects, each performing 9 types of actions. The number of synchronized frames collected for each subject across different actions is summarized in the Table6 below:

| Action type | Subj.2 | Subj.3 | Subj.4 | Subj.5 | Subj.7 | Subj.8 | Subj.9 | Subj.10 | Subj.11 |
|---|---|---|---|---|---|---|---|---|---|
| DJ1 | 100 | 86 | 109 | 111 | 114 | 98 | 92 | 104 | 90 |
| DJ2 | 106 | 87 | 104 | 110 | 103 | 106 | 94 | 100 | 85 |
| DJ3 | 102 | 88 | 112 | 109 | 99 | 111 | 92 | 87 | 91 |
| DJAsym1 | 119 | 86 | 91 | 119 | 102 | 123 | 93 | 88 | 118 |
| DJAsym4 | 118 | 86 | 93 | 110 | 103 | 140 | 91 | 89 | 101 |
| DJAsym5 | 113 | 90 | 100 | 108 | 90 | 133 | 96 | 91 | 80 |
| squats1 | 800 | 1184 | 1345 | 1623 | 1280 | 1320 | 1300 | 1288 | 1349 |
| squatsAsym1 | 900 | 1280 | 1338 | 1312 | 1340 | 1320 | 1350 | 1257 | 1414 |
| SitToStand1 | 860 | 1137 | 1399 | 1335 | 1370 | 1320 | 1440 | 1410 | 1328 |
| STSweakLeg1 | 1139 | 2187 | 1772 | 1630 | 2074 | 1860 | 1639 | 1383 | 1747 |
| walking1 | 158 | 132 | 130 | 139 | 132 | 129 | 135 | 125 | 135 |
| walking2 | 150 | 135 | 132 | 135 | 130 | 130 | 130 | 120 | 135 |
| walking3 | 153 | 138 | 139 | 140 | 133 | 135 | 137 | 130 | 138 |
| walkingTS1 | 184 | 180 | 143 | 150 | 150 | 197 | 144 | 147 | 163 |
| walkingTS2 | 170 | 170 | 140 | 145 | 157 | 149 | 140 | 150 | 157 |
| walkingTS4 | 184 | 155 | 140 | 145 | 170 | 154 | 137 | 142 | 160 |
| **Total counts** | 5356 | 7221 | 7287 | 7421 | 7547 | 7425 | 7110 | 6711 | 7291 |

Table 6: Number of valid synchronized frames for 9 subjects (Subj.2–Subj.11) across 16 action types (DJ = DownJump   STS = Sit-to-Stand).

A total of 51 reflective markers were attached to anatomical landmarks across the human body and recorded at 100 Hz. Markers on the lower limbs included the knee, ankle, calcaneus, 5th metatarsal, and toe, as well as segmental points on the shank and thigh (e.g., `r_knee`, `r_calc`, `r_toe`). Pelvic motion was captured using markers placed on the anterior and posterior superior iliac spines (`r.ASIS`, `L.ASIS`, `r.PSIS`, `L.PSIS`) together with hip joint centers (`R_HJC`, `L_HJC`, including regression-based estimates). Trunk movement was tracked by cervical (`C7`) and sternum markers (`R_Sternum`, `L_Sternum`). For the upper body, markers were placed on the shoulders, humerus, elbows (medial and lateral), forearms, and wrists (radius and ulna). This configuration enables reliable full-body kinematic reconstruction, providing accurate joint trajectories for subsequent inverse dynamics analysis. All the markers definition are shown in Table7.

The inverse dynamics results are stored in a structured text file containing 36 columns and 100 time frames. Each row corresponds to a time step and records the generalized joint forces computed from motion capture and ground reaction data. The variables include joint torques (e.g., hip flexion, adduction, and rotation moments for both sides; knee flexion moments; ankle plantar/dorsiflexion moments; subtalar and metatarsophalangeal joint moments), trunk and pelvis moments (lumbar extension, bending, rotation; pelvis tilt, list, and rotation moments), as well as translational forces acting on the pelvis. All values are expressed in SI units, with torques in Newton-meters (Nm) and forces in Newtons (N). This file thus provides the full set of generalized forces required for evaluating human inverse dynamics during motion. All the joints definition for the ground truth of joint torques are shown in Table8.

| Marker name | Anatomical location |
|---|---|
| r_knee, L_knee | Knee joint (right/left) |
| r_ankle, L_ankle | Ankle joint (right/left) |
| r_calc, L_calc | Calcaneus (heel bone) |
| r_5meta, L_5meta | Fifth metatarsal head |
| r_toe, L_toe | Toe marker |
| r_shank_*, L_shank_* | Shank segment points |
| r_thigh_*, L_thigh_* | Thigh segment points |
| r.ASIS, L.ASIS | Anterior superior iliac spine |
| r.PSIS, L.PSIS | Posterior superior iliac spine |
| R_HJC, L_HJC | Hip joint centers |
| C7 | 7th cervical vertebra |
| R_Shoulder, L_Shoulder | Shoulder joint |
| R_humerus, L_humerus | Humerus |
| R_elbow_med/lat, L_elbow_med/lat | Elbow (medial/lateral) |
| R_forearm, L_forearm | Forearm |
| R_wrist_radius/ulna, L_wrist_radius/ulna | Wrist (radius/ulna) |

Table 7: Marker naming convention and anatomical locations.

| Variable name | Description |
|---|---|
| pelvis_tilt_moment, pelvis_list_moment, pelvis_rotation_moment | Pelvis moments (tilt, list, rotation) |
| pelvis_tx_force, pelvis_ty_force, pelvis_tz_force | Pelvis translational forces (x,y,z) |
| hip_flexion_r/l_moment | Hip flexion/extension (right/left) |
| hip_adduction_r/l_moment | Hip adduction/abduction (right/left) |
| hip_rotation_r/l_moment | Hip internal/external rotation (right/left) |
| lumbar_extension_moment, lumbar_bending_moment, lumbar_rotation_moment | Lumbar spine moments |
| knee_angle_r/l_moment | Knee flexion/extension moments |
| ankle_angle_r/l_moment | Ankle plantar/dorsiflexion moments |
| elbow_flex_r/l_moment | Elbow flexion/extension moments |
| subtalar_angle_r/l_moment | Subtalar joint inversion/eversion moments |
| pro_sup_r/l_moment | Forearm pronation/supination moments |
| mtp_angle_r/l_moment | Metatarsophalangeal (toe) joint moments |
| arm_flex/add/rot_r/l_moment | Shoulder moments (flexion, adduction, rotation) |

Table 8: Joint positions defined in the torque ground truth.

### A.3 EFFECT OF WINDOW SIZE

In the main experiments, we adopted a temporal window size of 13 frames for torque prediction using the Transformer backbone. To assess the robustness of this design choice, we additionally evaluated the model under different window sizes, including 9 and 17 frames. As shown in Table 9, the overall performance remains consistent across different settings, with a slight degradation for very short windows (9 frames), likely due to insufficient temporal context. These results suggest that a window size of 13 frames provides a good trade-off between model accuracy and computational efficiency.

| Window size | 9 | 13 (default) | 17 |
|---|---|---|---|
| mPJE $\downarrow$ | 1.841 | **1.7612** | 1.7662 |

Table 9: Performance of Transformer models under different temporal window sizes.

### A.4 THE USE OF LARGE LANGUAGE MODELS

Large Language Models (LLMs) provide valuable assistance in scientific writing by automating both technical formatting and linguistic refinement. In our workflow, LLMs were employed to generate and optimize LaTeX table styles, ensuring that the presentation of experimental results adheres to common academic standards such as clean alignment, minimal use of grid lines, and the adoption of the booktabs package. In addition, LLMs were used to refine the manuscript text, improving readability, grammatical accuracy, and overall fluency so that the writing conforms more closely to native academic English. This dual use of LLMs—technical support for LaTeX and stylistic support for writing—helped streamline the preparation of this paper while maintaining both clarity and professionalism. We ensured that no content was generated by LLMs beyond formatting and language refinement, to maintain scientific rigor.

