# OpenReview forum: "Towards Human Inverse Dynamics from Real Images: A Dataset and Benchmark for Joint Torque Estimation"
_ICLR.cc/2026/Conference — ICLR 2026 Conference Withdrawn Submission_

### Official Review · Reviewer_bxV7 · 2025-10-23

**Soundness:** 2
**Presentation:** 2
**Contribution:** 3
**Rating:** 4
**Confidence:** 4

**Summary:**

This paper introduces VID, a new dataset for predicting human joint torques directly from real RGB images, an important step towards vision-based biomechanics. VID includes 63k synchronized frames with visual, kinematic, and dynamic annotations derived and refined from open-source datasets using OpenSim. The authors further propose VID-Network, a CNN–Transformer hybrid designed to infer joint torques from monocular images, using auxiliary modules for pose estimation and marker regression. The model achieves significantly lower mean per-joint torque error (mPJE) than previous approaches such as ImDy and Dino. The work establishes a first benchmark for this task and provides baseline results and evaluation protocols.

**Strengths:**

1. VID is the first dataset specifically built for predicting joint torques from real human images, bridging an important gap between visual human motion understanding and biomechanics.

2. The VID-Network provides a solid baseline architecture and introduces a well-structured benchmark with joint-level and action-level evaluation criteria.


3. This dataset can catalyze new research directions in vision-based biomechanics, rehabilitation, and sports analysis.

**Weaknesses:**

1. The dataset details are incomplete. Key aspects such as camera placement, rest intervals between actions, environmental setup, and synchronization validation procedures are missing.

2. The paper mentions smoothing using a Savitzky–Golay filter but does not quantify or visualize its effect on reducing noise. A before–after illustration would strengthen data reliability claims.

3. The dataset scale is given in frames (63,369 f) while others are in hours, this should be unified for direct comparison.

4. The baseline relies solely on CNN features. Given the rise of vision transformers and video foundation models, the benchmarking would be more convincing if alternative backbones (e.g., ViT, Video Swin, Mamba-based architectures) were compared or discussed.

5. The pose estimation module is implemented as a simple CNN head, whereas stronger pretrained pose estimators (e.g., OpenPose, MMPose) could be easily integrated. The paper should clarify whether such alternatives were tested and how they affect torque prediction.

6. The chosen loss weights (λ₁ = λ₂ = 0.5) lack justification; additional ablations on these hyperparameters are needed.  Some notations (e.g., Mass_sub in Eq. 8) are undefined or ambiguously described.

7. The results mainly focus on numerical improvements; qualitative visualizations are scarce. Only one qualitative example is shown—additional samples, including failure cases, would reveal limitations and robustness.

8. Recent efforts in video-based muscle activation estimation (e.g., [1] and [2]) are relevant and should be discussed to position this work properly within the field of visual biomechanics.

[1]  Peng, Kunyu, et al. "Towards video-based activated muscle group estimation in the wild." Proceedings of the 32nd ACM International Conference on Multimedia. 2024.

[2] Schneider, David, et al. "Muscles in time: Learning to understand human motion in-depth by simulating muscle activations." Advances in Neural Information Processing Systems 37 (2024): 67251-67281.

9. While VID introduces a valuable benchmark, the evaluation lacks analysis on cross-subject generalization or domain transfer, both crucial for real-world deployment.

**Questions:**

1. Could the authors elaborate on the camera setup during data collection? Where was the camera positioned relative to the subject (distance, height, angle)? Was it static or moving? Were lighting and background conditions controlled or varied?

2. How were rest intervals between two consecutive actions determined to avoid fatigue effects or pose drift? Were all participants given the same duration of rest? Could uneven rest times affect the naturalness of motion or torque distribution?

3. How was data synchronization validated between the kinematic, dynamic, and visual streams? Was any quantitative synchronization error analysis performed?

4. The paper mentions using a Savitzky–Golay filter for smoothing kinematic trajectories. Could the authors quantitatively demonstrate the improvement in signal quality (e.g., noise variance reduction)? Can they visualize before–after filtering examples to justify the filtering window and order parameters? What is the estimated label noise ratio before and after filtering?

5. Table 1 compares datasets using different scales (hours vs. frames). Could the authors convert the total frame count into equivalent recording time to make comparisons directly interpretable?

6. The baseline employs a CNN feature extractor. Have the authors considered transformer-based or video foundation models (e.g., Video Swin, TimeSformer, Mamba-based backbones)? If not, could they discuss the expected benefits or trade-offs of using such architectures? Would these alternatives improve temporal modeling and generalization across unseen motions?

7. For the pose estimation module, have the authors tested pretrained 2D/3D pose estimators such as OpenPose, MMPose, or HMR? If not, why was a self-trained CNN head chosen over stronger pretrained options? Could using such models as frozen priors enhance pose accuracy and consequently torque estimation?

8. The loss function uses λ₁ = λ₂ = 0.5 without justification. How were these values selected? Were different combinations (e.g., λ₁:λ₂ = 0.3:0.7 or 0.7:0.3) tested? Could the authors include an ablation on loss weights to demonstrate sensitivity and optimal trade-off?

9. Equation (8) introduces Mass_sub, but its definition is ambiguous. Does this refer to subject-specific body mass measured during data collection, or a normalized value? Is it constant per subject or variable per frame?

10. The paper shows only one qualitative visualization. Could the authors include more qualitative examples covering multiple actions and joints? Have they analyzed failure cases (e.g., occlusions, fast motion) to understand model weaknesses? Visual comparisons with other methods (e.g., ImDy, Dino) would be informative.

11. The paper overlooks recent visual biomechanics studies (e.g., Peng et al., 2024; Schneider et al., 2024). How does VID differ conceptually from these video-based muscle activation datasets? Could the authors clarify whether VID could be extended to predict muscle activation or EMG proxies in the future?

12. The evaluation focuses on within-dataset testing. Have the authors conducted any cross-subject or cross-action generalization experiments? How does the model perform when trained on certain subjects but tested on unseen ones? Could the authors discuss the domain transferability of VID-trained models to real-world or in-the-wild footage?

---

### Official Review · Reviewer_EUmX · 2025-10-27

**Soundness:** 2
**Presentation:** 2
**Contribution:** 2
**Rating:** 2
**Confidence:** 4

**Summary:**

The paper addresses the needs of large-scale training data for the task of human inverse dynamics estimation.
The proposed dataset VID contains synchronized data in multiple different modalities: images, kinematics and dynamics.
To address the limitation of sensory devices such as EMGs to real-world usage, the paper proposes using OpenSim as the alternative solution for dynamics capture.
Based on the collected dataset, the paper also uses a neural model, VID Network, to learn human dynamics using images as input, and the model is trained with the collected data from OpenSim as ground truths.
Quantitative results show performance gains of VID Network on the proposed VID dataset, compared to two other baselines Dino and ImDy.

**Strengths:**

- The field is indeed in need of high-quality dynamics data, and the idea of deriving the data directly from the kinematics evidence via OpenSim is appreciated.
- The paper provides a first benchmark for human inverse dynamics from images on the proposed VID dataset.
- The proposed VID Network is well-designed and achieves better performance than the baseline methods on VID dataset.

**Weaknesses:**

1. On the VID dataset.
- There is currently no experiment or comparative study to verify the accuracy of OpenSim data.
Since OpenSim is also a simulation with its own simulated biases, the results might be implausible and not suitable to be used as ground truths.
There should be a comparison between data collected from OpenSim and data collected from sensors such as EMGs or force plates.
- The details about the synchronization process and the criteria for a frame to be considered as valid (Tab. 6) are not demonstrated. Given that the VID dataset is derived from OpenCap dataset, the evidence of low-quality examples from OpenCap that need processing and synchronization are not presented.

2. On the evaluation metrics and results.
- Mean per-joint error and its variations like action-wise measurements have been used by the community for a long time, I do not think claiming novelty here is suitable.
- It is unclear whether the two baselines are also exposed to the new VID data. If not, it is not fair to compare a method that is directly train on VID dataset to those that are not. How about evaluating on a different dataset without fine-tuning?
- The example in Fig. 3 is not a good visualization since it does not show the improvement of VID very clearly. Maybe zooming to the period of interest from frame 45 to frame 70 would help.

3. On the manuscript presentation.
- Confusing notation in Eq. 2, where $m*t$ makes it seems like the number of joints is scaled linearly w.r.t time step t.
- In Tab. 1, comparing number of hours $h$ from other datasets to number of frames $f$ in VID causing confusions.
- In line 304 to 311, the meaning of dimensions $N$ and $d$ are unknown and cannot be observed in Fig. 2. Given TorqueInferNet is the most important network for dynamics estimation, the current writing makes it hard to follow.

**Questions:**

- Regarding the Finite Difference calculation in line192, should there be an equation label here?
And what is the current joint rotation representation being used in VID? And is $v_i$ the joint rotation velocity or linear velocity?
- On the argument in line 057-058, does OpenSim’s data also suffer the same discrepancies since it is also a simulation method?
- In Fig.2, why the joints torque tensor only has shape $B\times11\times1$? Does this mean that VID only consider 1D rotational torques?

---

### Official Review · Reviewer_W3hs · 2025-10-30

**Soundness:** 2
**Presentation:** 2
**Contribution:** 2
**Rating:** 4
**Confidence:** 4

**Summary:**

The authors proposed the VID as a vision-based human inverse dynamics dataset. VIDNet is further designed to directly regress human joint torque from images. Experiments show the efficacy of the proposed pipeline.

**Strengths:**

- It addresses a key barrier in biomechanics by attempting torque estimation from vision alone, which could enable analyses outside laboratory settings where force plates and marker-based systems are impractical.

- The data curation effort is appreciated. VID assembles a dataset aligning images, kinematic trajectories, and joint torques, with manual synchronization and outlier handling. Includes multiple actions and anthropometrics, which are potentially valuable for supervised learning of dynamics from images.

- A coherent multi-task learning framework is proposed and shown to be superior to baselines.

- Comprehensive analyses are provided by demonstrating overall, joint-wise, and action-wise results, plus an ablation across the main modules, providing useful insight into which components drive performance.

**Weaknesses:**

- More analyses on failure cases would be appreciated.

  - How does prolonged occlusion of a limb (e.g., by furniture or another person) impact the dynamics estimation?

  - What happens when the model encounters highly dynamic or contorted poses (e.g., gymnastics, breakdancing) that are far outside the distribution of the training data?

  - The performance under non-ideal imaging conditions, common in real-world applications, should be investigated.

- More detailed analyses on cross-subject and cross-action evaluation would be helpful.

  - A more detailed cross-subject analysis is needed. Instead of just reporting a single aggregate error metric, it would be helpful to analyze if the model's performance degrades predictably for subjects with specific biomechanical characteristics (e.g., higher/lower body mass index, significantly different height, or unique gait patterns) not well-represented in the training set. Does the model learn a subject-invariant representation of dynamics, or does it subtly overfit to the anthropometrics of the training subjects?

  - Cross-Action Evaluation: This is a pivotal point. The manuscript would benefit greatly from a "leave-one-action-out" or similar cross-action validation scheme. For example, train the model on data from actions A, B, and C, and test it on action D. This experiment directly tests the model's ability to understand the fundamental physics of human motion, rather than memorizing action-specific patterns. Analyzing which action transitions work well and which fail would be extremely revealing about what the model has truly learned. Does knowledge of "walking" help in estimating the dynamics of "running," or are they treated as completely separate domains?

- It will be better if the model size comparison could be included.

**Questions:**

- The source data of VID seems to be a subset of AddBiomechanics. Are there any differences in the processed dynamics data?

- Are there any data quality analyses for VID, like the Hicks threshold?

---

### Official Review · Reviewer_z9x8 · 2025-11-01

**Soundness:** 3
**Presentation:** 2
**Contribution:** 2
**Rating:** 2
**Confidence:** 4

**Summary:**

The paper proposes a vision-only inverse dynamics method for predicting joint torques directly from single RGB images, without any additional modalities. According to the authors, the contribution rests on two parts: a dataset/benchmark and a baseline model. The dataset, VID, contains 63,369 synchronized frames (RGB, kinematics, dynamics) derived from public sources and processed with OpenSim; it includes a three-level benchmark (overall, joint-specific, action-specific). The baseline, VID-Network, combines a CNN backbone, a spatial probabilistic head with soft-argmax, a marker regressor, and a temporal TorqueInferNet (Transformer, 13-frame window). Experiments (2×A100, 8:2 split) use mPJE (Nm/kg) and report gains over ImDy and Dino on most criteria.

**Strengths:**

1.	Novel dataset. A new, synchronized RGB–kinematics–dynamics dataset tailored to torque estimation.
2.	Novel task. A benchmark focusing on torque prediction from real images without force plates or sEMG.
3.	Problem & dataset framing. The task is well-motivated; the dataset specification and tables are clear.
4.	Method clarity. The baseline pipeline (pose pretrain, spatial head with soft-argmax, marker regressor, temporal regressor) is described with equations and ablations.

**Weaknesses:**

1.	Motivation for the single-image setting. Please clarify why estimating joint torque from single images (vs. short videos) is the target setting, given that temporal cues and physics signals are often more informative than a single frame.
2.	Baseline coverage (input-matched). The benchmark lacks clear RGB-input baselines. Several viable routes should be considered or clarified:
•	RGB → torque (end-to-end, RGB-only)
•	RGB → 2D keypoints → torque (RGB-only)
•	RGB → 2D → 3D skeleton → torque
•	Inverse dynamics from kinematics (e.g., Newton–Euler/OpenSim using predicted joint angles/velocities/accelerations)
•	Imitation learning from kinematic motion
3.	Ground-truth dynamics & external forces. Without force plates, how are GRFs/external loads handled in OpenSim when generating torque labels? The derivation pipeline (model choice, scaling, constraints) needs more detail for label validity.
4.	Metric breadth. Only mPJE (Nm/kg) is reported. Consider adding more metrics; action-specific results are useful, but statistical significance remains unclear.
5.	Non-matched baselines. ImDy/DINO typically assume motion/marker inputs, whereas the proposed baseline uses RGB. How were these adapted (e.g., via predicted markers)? Otherwise the comparison may not be apples-to-apples.
6.	What does “DINO” refer to here? Please clarify the exact baseline variant, inputs, and training protocol for DINO in this setting.
7.	Dataset scale & generalization. The dataset is relatively small; cross-subject and in-the-wild evaluations are listed as future work, but current generalization remains untested.
8.	Definition clarity. The distinction between marker positions and joint positions is unclear—please formalize both and how each is used in training/evaluation.
9.	Notation/dimensions.
•	Lines 256–257: what is the dimension of the SPF feature?
•	Lines 366–367: what is the dimension of \hat{J} (predicted joints)? Please also define the full formulation around these variables.

**Questions:**

1.	Why prioritize a single-image setting over a short-video setup? Any results with temporal inputs for comparison?
2.	Which RGB-input baselines did you implement? If prior methods require motion/markers, how exactly were they adapted (e.g., predicted markers)?
3.	How were GRFs/external forces modeled in OpenSim when generating torque labels without plates?
4.	Will you report additional metrics alongside mPJE?
5.	What, precisely, is the DINO baseline here (variant, inputs, training recipe)?
6.	Which pretraining datasets were used, and how are camera intrinsics/extrinsics handled to ensure metric consistency for soft-argmax outputs?
7.	Please clarify the definitions of markers vs joints, and specify the dimensions of SPF (lines 256–257) and \hat{J} (lines 366–367), with complete equations.

---

### Note · Authors · 2025-11-25

I have read and agree with the venue's withdrawal policy on behalf of myself and my co-authors.